# Biological Characteristics of Feline Calicivirus Epidemic Strains in China and Screening of Broad-Spectrum Protective Vaccine Strains

**DOI:** 10.3390/vaccines11121858

**Published:** 2023-12-15

**Authors:** Longlong Cao, Jian Liu, Yongfan Li, Denglong Xie, Quanhui Yan, Qiuyan Li, Yiran Cao, Wenxin Du, Jiakang Li, Zijun Ye, Dengyuan Zhou, Chao Kang, Shengbo Cao

**Affiliations:** 1Wuhan Keqian Biology Co., Ltd., Wuhan 430070, China; caolong5955@163.com (L.C.); lyf1305344934@163.com (Y.L.); yanqh@webmail.hzau.edu.cn (Q.Y.); lqylqy6@webmail.hzau.edu.cn (Q.L.); adrena069@163.com (W.D.); 18408213507@163.com (J.L.); junbiomedical@163.com (Z.Y.); zhoudy6@163.com (D.Z.); 2National Key Laboratory of Agricultural Microbiology, College of Veterinary Medicine, Huazhong Agricultural University, Wuhan 430070, China; 3Shanghai Animal Disease Prevention and Control Center, Shanghai 201103, China; jianjian115@sohu.com; 4Zhejiang Hisun Animal Healthcare Products Co., Ltd., Hangzhou 311400, China; denglong.xie@hisunah.com; 5Department of Life Science, Imperial College London, London SW7 2AZ, UK; yc4122@ic.ac.uk

**Keywords:** feline calicivirus, genetic diversity, inactivated vaccine, neutralizing antibodies, extensive cross-protection

## Abstract

Feline calicivirus (FCV) is one of the most important pathogens causing upper respiratory tract diseases in cats, posing a serious health threat to these animals. At present, FCV is mainly prevented through vaccination, but the protective efficacy of vaccines in China is limited. In this study, based on the differences in capsid proteins of isolates from different regions in China, as reported in our previous studies, seven representative FCV epidemic strains were selected and tested for their viral titers, virulence, immunogenicity, and extensive cross-protection. Subsequently, vaccine strains were selected to prepare inactivated vaccines. The whole-genome sequencing and analysis results showed that these seven representative FCV strains and 144 reference strains fell into five groups (A, B, C, D, and E). The strains isolated in China mainly fall into groups C and D, exhibiting regional characteristics. These Chinese isolates had a distant evolutionary relationship and low homology with the current FCV-255 vaccine strain. The screened FCV-HB7 and FCV-HB10 strains displayed desirable in vitro culture characteristics, with the highest virus proliferation titers (10^9.5^ TCID_50_/mL) at 36 h post inoculation at a dose of 0.01 MOI. All five cats infected intranasally with FCV-HB7 or FCV-HB10 strains showed obvious clinical symptoms of FCV. The symptoms of cats infected with the FCV-HB7 strain were more severe than those infected with the FCV-HB10 strain. Both the single-strain inactivated immunization and combined bivalent inactivated vaccine immunization of FCV-HB7 and FCV-HB10 induced high neutralizing antibody titers in five cats immunized. Moreover, bivalent inactivated vaccine immunization protected cats from FCV-HB7 and FCV-HB10 strains. The cross-neutralizing antibody titer against seven representative FCV epidemic strains achieved by combined bivalent inactivated vaccine immunization was higher than that achieved by single-strain immunization, which was much higher than that achieved by commercial vaccine FCV-255 strain immunization. The above results suggest that the FCV-HB7 and FCV-HB10 strains screened in this study have great potential to become vaccine strains with broad-spectrum protective efficacy. However, their immune protective efficacy needs to be further verified by multiple methods before clinical application.

## 1. Introduction

*Feline calicivirus* (FCV) belongs to the family *Caliciviridae* and genus *Vesivirus*, is one of the main pathogens of upper respiratory tract diseases in felines [1,2], and it mainly causes symptoms such as oral ulcers and respiratory tracheitis in felines. FCV can induce severe disease in domestic cats and large wild felines such as tigers and lions [3,4]. FCV is a non-enveloped single-stranded positive-sense non-segmented RNA virus with a diameter of 35 to 40 nm [5,6]. The nucleocapsid has an icosahedral symmetry structure and a genome of approximately 7.7 kb [7,8]. The genome contains three open reading frames (*ORFs*), of which *ORF1* encodes non-structural proteins such as P30; *ORF2* encodes the capsid protein precursor; and *ORF3* encodes the minor structural protein VP2 [9,10]. The FCV capsid protein precursor is further processed to form the main capsid protein VP1 and the leader of the capsid (LC). The main capsid protein VP1 is divided into six regions (A–F) [10]. Compared with other regions, C and E are highly variable [6], containing most of virus-neutralizing epitope regions [11]. The genetic variability in the C, D, and E regions will affect the immunogenicity of FCV, which is responsible for the low-serum cross-reactivity of different FCV strains [4]. Currently, there is only one serotype of FCV. The RNA polymerase of FCV lacks a sequence correction function with low fidelity, making the FCV genome prone to mutations during virus replication [6]. This mutation-prone virus replication mechanism allows the virus to respond more quickly and produce multiple genotypes under immune pressure, making the evasion of the virus from immune clearance easier [12]. The resulting immune evasion is one of the main reasons for FCV vaccine failure [13,14]. Recently, the clinical symptoms of FCV infection evolved from typical upper respiratory tract and oral ulcer symptoms to lower respiratory tract diseases (such as severe pneumonia) [15]. The recently emerged highly pathogenic FCVs, such as virulent systemic calicivirus (VS-FCV), can cause a systemic viremic disease with fever, oral ulcers, skin ulcers, jaundice, and liver and other tissue necrosis, with a mortality rate as high as 67% [16,17].

FCV causes epidemics worldwide, and FCV-infected cats continuously shed the virus for months or years [18]. Currently, the primary measure for FCV prevention is vaccination [19]. FCV vaccines are divided into attenuated live vaccines and inactivated vaccines [20,21]. Most of the FCV vaccines used in China are inactivated FCV vaccines [22]. The variability of FCV antigens results in the poor efficacy of the vaccines and the failure to completely overcome epidemic strain infection and prevent virus transmission, achieving only the alleviation of some clinical symptoms [23]. It is reported that existing vaccines have a certain protective effect against homologous FCV strains, but due to the large antigenic differences between FCV strains, the neutralizing titers against heterologous FCV strains are low [24,25]. Cats with a history of immunization are still susceptible to new epidemic strains [26]. The persistent high mutability of FCV epidemic strains has greatly increased the clinical demand for new FCV vaccines [27]. Therefore, the diversity of FCV genotypes should be taken into account in the development of new FCV vaccines [28]. In this study, based on the differences in capsid proteins of isolates from different regions in China, as reported in our previous studies [29], seven representative FCV epidemic strains were selected and tested for their viral titers, virulence, immunogenicity, and extensive cross-protection. Subsequently, vaccine strains were selected to prepare inactivated vaccines. Based on the screening, a multivalent FCV inactivated vaccine was prepared to improve the cross-protection rate of the FCV vaccine against epidemic strains in China.

## 2. Methods

### 2.1. Cells and Virus

F81 cells were passaged, identified, and preserved by our laboratory. These cells were propagated in Dulbecco’s modified Eagle’s medium (DMEM; Gibco, Waltham, MA, USA) supplemented with 10% fetal bovine serum (FBS; Gibco, Waltham, MA, USA), 100 U/mL penicillin, and 100 μg/mL streptomycin and incubated at 37 °C with 5% CO_2_. The feline caliciviruses FCV-FJ1, FCV-AH3, FCV-JL18, FCV-SH192, FCV-HB260, FCV-HB7, and FCV-HB10 (GenBank accession numbers OR645479-OR645485) were derived from patients with typical FCV symptoms (ORD and conjunctivitis) in cats, isolated by our laboratory after collecting nasal, ocular, and oral samples using sterile cotton swabs. The identification results have been published in previous reports [29].

### 2.2. Detection of FCV Whole-Genome Sequence Mutation

The primers were designed based on the NCBI sequences for the amplification of the FCV whole-genome sequence (Table 1), and these primers exhibited good specificity. RT-PCR was performed using a commercially available one-step kit (Takara Biomedical Technology, Beijing, China). The RT-PCR amplification products were TA-cloned and sequenced by Sangon Biotechnology Co., Ltd. (Shanghai, China), and the sequences were spliced using DNAStar (Madison, WI, USA).

### 2.3. Phylogenetic Analyses

The whole-genome sequences of typical FCV strains were obtained from the National Center for Biotechnology Information (NCBI) database. The FCV isolate sequences in this study were submitted to GenBank. Whole-genome sequences were aligned using Mega X at the nucleotide level (11.0.13, Pennsylvania State University, State College, PA, USA) and Mega X and DNAStar (17.1.1, Madison, WI, USA) at the amino acid level (Pennsylvania State University, USA) with manual adjustments. The nucleotide sequences and amino acid sequences were compared between the isolated strains and reference strains. A phylogenic tree was constructed using the Maximum Likelihood (ML) method with 1000 bootstrap replicates.

### 2.4. FCV Virus Titer Assay

F81 cells infected with 7 FCV strains isolated and identified in our laboratory were harvested and observed when cytopathic effect (CPE) lesions were complete under a microscope, and the virus titer was evaluated via a 50% tissue culture infective dose (TCID_50_) assay. The F81 cells were inoculated into 96-well plates and incubated in DMEM containing 5% NBS. The virus samples were 10-fold serially diluted with DMEM, inoculated into F81 cells, and cultured at 37 °C for 48–72 h. CPE was observed after F81 cell infection by the virus, and TCID_50_ was calculated using the Reed–Muench method.

### 2.5. Immunogenicity Assay of FCV Strains

#### 2.5.1. Purification and Inactivation of FCV Virus

Each FCV strain culture solution was purified and concentrated by ultracentrifugation at 130,000× *g* for 2 h at 4 °C. The supernatant was discarded, and the precipitate was resuspended in 500 μL sterile PBS, which was added into a gradient sucrose solution (10, 20, 30% sucrose) and centrifuged at 130,000× *g* for 2 h at 4 °C. Finally, the FCV strains were harvested and inactivated by adding β-propiolactone at the ratio of 0.06% (β-propiolactone:strain solution) for FCV antigen preparation. The specific method involves adding β-propiolactone and shaking at 4 °C for 24 h. After that, the β-propiolactone is hydrolyzed at 37 °C for 2 h. Once inactivation is complete, the antigen is inoculated into the F81 cell culture medium covering the T25 monolayer at a volume ratio of 1:100. Daily observation of the lesions is conducted. If no lesions are observed, harvesting is performed by freezing and thawing on the 6th day. Blind passage is then performed for 3 generations using the same inoculation method. If no lesions are observed in the 3 generations, the inactivation process is considered successful. The inactivated antigen and MONTANIDE™ GEL 02 PR (Seppic, Paris, France) adjuvant are mixed at a volume ratio of 9:1 to prepare the vaccine.

#### 2.5.2. Antibody Titer Determination using Virus Neutralization Test

The feline serum samples were centrifuged (3000× *g*, 10 min) and inactivated (56 °C, 30 min). In the virus neutralization test (VN), the serum was diluted at the ratios of 1:2 to 1:4096 in a 96-well microtiter plate, and added with the virus (10^2^ TCID_50_/well). After 1 h of reaction of the serum and the virus, the F81 cells were added into 96-well plates containing both serum and virus (2.0 × 10^4^/well), and cultured at 37 °C in 5% CO_2_ for 4–5 days. The neutralizing antibody titers were calculated according to the cytopathic effect using the Reed–Muench method.

#### 2.5.3. Immunogenicity Assay of FCV Strains

We conducted a preliminary immunogenicity assay with 7 FCV strains using SPF mice as hosts. Here, 40 mice were randomly divided into 8 groups, with 5 mice in each group. Each of the 7 inactivated FCV strains was injected into 5 mice in the same group at a dose of 10^8.5^ TCID_50_/mouse (50 μL strain solution mixed with adjuvant at the ratio of 1:1). Group 8 served as the negative control group, which was immunized with the same dose of PBS and adjuvant. An enhanced immunization was performed on the 21st day of the experiment, and the serum of mice in each group was collected on the 21st day after the enhanced immunization. The neutralizing antibody titer of each group was determined. The strain with the highest virus proliferation and highest neutralizing antibody titer was screened and used as the vaccine strain.

### 2.6. Determination of Dynamic Growth Curve of FCV Vaccine Strains

The selected vaccine strains were inoculated into F81 cells at dose of 0.1 MOI, 0.01 MOI, 0.001 MOI, and 0.0001 MOI, respectively. The cell culture was collected at 12 h, 24 h, 36 h, and 48 h post inoculation (hpi). The TCID_50_ value was measured using the Reed–Muench method. The growth of FCV vaccine strains in vitro was investigated.

### 2.7. Pathogenicity Determination of FCV-HB7 and FCV-HB10 Strains in Cats

All of the cats used in this study were maintained in compliance with the recommendations in the Regulations for the Administration of Affairs Concerning Experimental Animals made by the Ministry of Science and Technology of China. The experiments were performed using protocols that were approved by the Scientific Ethics Committee of Huazhong Agricultural University (permit number: HZAUCA-2023-0030). Researchers are responsible for complying with all applicable laws, rules, and regulations regarding animal welfare. Fifteen healthy British shorthair cats (FCV neutralizing antibody <1:2 and FCV antigen negative), aged 8–12 weeks, were randomly divided into three groups (FCV-HB7 group, FCV-HB10 group, and control group), with five cats in each group. Five cats per group were raised in an animal house (3.2 m × 3.2 m). Each cat in the FCV-HB7 group was inoculated intranasally with 1 mL (10^9.0^ TCID_50_/mL) FCV-HB7 isolate; each cat in the FCV-HB10 group was inoculated intranasally with 1 mL (10^9.0^ TCID_50_/mL) FCV-HB10 isolate (the use of high-virus titers allows for the establishment of an animal experimental model to better evaluate the protective effect of subsequent vaccine immunization), and each cat in the control group was inoculated intranasally with an equal volume of DMEM. The clinical signs displayed by the cats were observed and assessed daily until 14 days post inoculation (dpi) [15,30]. Referring to the *European Pharmacopoeia* and relevant literature, the clinical symptom scoring system was formulated, and the clinical symptoms of cats were scored daily until 14 dpi, with detailed scores shown in Table 2 [15,30]. The rectal body temperature and body weight of the cats were monitored daily throughout the experiment. We performed double-blind (participant and rater) assessments to avoid significant bias in clinical scoring. On 14 dpi, all of the cats were euthanized according to the methods for the Euthanasia of Dogs and Cats suggested by the World Society for the Protection of Animals (https://www.rspca.org.uk/, accessed on 8 March 2021). The lung tissues and nasal turbinates were subjected to histopathological assay.

### 2.8. Cat Protective Efficacy Assay

To further evaluate the immunogenicity of FCV vaccine strains, 35 healthy British Shorthair cats (FCV neutralizing antibody <1:2 and FCV antigen negative) aged 8–12 weeks were randomly divided into 7 groups with 5 cats per group, and each group was raised separately in an animal house (3.2 m × 3.2 m). The experiments were performed using protocols that were approved by the Scientific Ethics Committee of Huazhong Agricultural University (permit number: HZAUCA-2023-0030). Researchers are responsible for complying with all applicable laws, rules, and regulations regarding animal welfare. Each cat in group 1 was immunized with the inactivated FCV-HB7 strain at a dose of 10^8.5^ TCID_50_; each cat in group 2 was immunized with the inactivated FCV-HB10 strain at a dose of 10^8.5^ TCID_50_; each cat in group 3 and 4 was immunized with inactivated 10^8.5^ TCID_50_ of FCV-HB7 and FCV-HB10 strains; each cat in group 5 was immunized with the inactivated FCV-255 strain (commercial vaccine) according to the dosage specified by commercial vaccine instructions; each cat in group 6 and 7 (non-immunization control groups) was immunized with the same amount of DMEM. The cats in each group received a booster vaccination on the 21st day after the initial vaccination.

Considering animal welfare, only 4 groups were selected for FCV strain challenge. Serum samples were collected on the 14th and 35th days after the initial vaccination to track neutralizing antibodies. On day 21 after the second immunization, the cats in groups 3 and 6 were intranasally infected with the 10^9.0^ TCID_50_/mL FCV-HB7 strain, and cats in groups 4 and 7 were intranasally infected with the 10^9.0^ TCID_50_/mL FCV-HB10 strain. After the FCV strain challenge, clinical symptoms were scored every day (see Table 2), and rectal body temperature and body weight were measured for 14 consecutive days. Eye and nasopharyngeal swabs were collected every day and placed in sterile saline, and the TCID_50_ method was used to detect virus shedding in the swabs. At 14 dpi, all of the cats were euthanized according to the methods for the Euthanasia of Dogs and Cats suggested by the World Society for the Protection of Animals (https://www.rspca.org.uk/). Pathological lesions and the viral load of lung tissues and nasal turbinates were analyzed. Finally, the serum of one cat from groups 1, 2, 3, and 5 was randomly selected on day 35 after the initial immunization for determining in vitro cross-neutralizing titers against FCV-representative strains.

### 2.9. Hematoxylin and Eosin (H&E) Staining

The samples of lung tissues and nasal turbinates of the tested cats were randomly collected, fixed in 10% neutral formalin, embedded in paraffin, cut into 3 μm thick sections, and stained with hematoxylin and eosin (H&E). Photos were taken under an optical microscope.

### 2.10. Statistical Analysis

All data were analyzed using GraphPad Prism 8.0 and presented as mean ± standard deviation (SD). Comparisons were performed using a *t*-test or two-way ANOVA.

## 3. Results

### 3.1. Homology Analysis of FCV Whole-Genome Sequence

Through sequencing and splicing, the whole-genome sequences of seven representative FCV isolates from different regions in China were obtained (Table 3). The whole-genome nucleotide sequence homology among the seven representative isolates ranged from 77.29 to 90.95%. The nucleotide sequence homology between seven isolates from China and 144 reference strains with global origins ranged from 76.09 to 96.01%. Overall, all seven isolates exhibited whole-genome nucleotide mutations compared with the FCV reference strains, especially compared with the FCV 255 commercial vaccine strain (GenBank accession number: KM111170.1), with a nucleotide sequence homology of only 77.37~80.60%. The homology analysis results showed that the amino acid homology of *ORF1, ORF2*, and *ORF3* between the seven isolate strains and the FCV-255 vaccine strain was 86.91–91.16%, 83.53–88.04%, and 83.02–95.28%, respectively (Table 4).

### 3.2. Phylogenetic Analyses of FCV Isolates

The whole-genome nucleotide sequence-based phylogenetic tree revealed five main groups, namely, A, B, C, D, and E. Among them, group A mainly included strains isolated from the United States and Japan; group B included isolates from multiple countries, but mainly from the United States; groups C and D mainly included isolates from China, and group E mainly included isolates from Australia. According to the evolutionary relationship, all seven of our isolates were genetically distant from vaccine strain 255 (Figure 1). Of these seven isolates, four belonged to group C and three belonged to group D, whereas the FCV-255 vaccine strain currently used in China belonged to Group B.

### 3.3. Determination of Virus Titer and Immunogenicity of FCV Strains

Complete CPE was observed under the microscope at 24–36 h after inoculating seven FCV strains into F81 cells. Subsequently, F81 cell cultures were collected, and the TCID_50_ was measured. The results showed that the FCV-HB7 strain had the highest virus titer (10^9.5^ TCID_50_/mL), followed by the FCV-HB10 strain (10^9.4^ TCID_50_/mL). The FCV-HB260 strain had the lowest virus titer, which was 10^8.0^ TCID_50_/mL (Figure 2A).

The results showed that the FCV-HB7 strain induced the highest neutralizing antibody titer after immunization, with an average of 1:2502, followed by the FCV-HB10 strain (1:2098), and the FCV-JL18 strain exhibited the lowest neutralizing antibody titer, with an average of 1:106 (Figure 2B). The FCV-HB7 strain and FCV-HB10 strain were selected as candidate FCV vaccine strains.

### 3.4. Growth Kinetics of FCV-Representative Strains

The results showed that the inoculations of different strains and different doses resulted in different replication properties. The lower the dose, the slower the virus replication. The FCV-HB7 strain and FCV-HB10 strain exhibited the highest virus titer at 36 hpi at an inoculation dose of 0.01 MOI, both of which were 10^9.5^ TCID_50_/mL. It was higher than the maximum virus titer of FCV-HB7 strain and FCV-HB10 strain at an inoculation dose of 0.1 MOI at different time points, which was 10^9.1^ TCID_50_/mL and 10^9.0^ TCID_50_/mL, respectively (Figure 3).

### 3.5. Pathogenicity of FCV-HB7 and FCV-HB10 in Cats

Cats infected with the FCV-HB7 strain or FCV-HB10 strain showed obvious clinical symptoms such as abnormal body temperature, weight loss, increased eye and nose secreta, conjunctivitis, oral ulcers, and even death. The clinical symptom scores within 14 dpi are shown in Figure 4B. The symptoms of cats challenged with the FCV-HB7 strain were more severe than those of the cats challenged with the FCV-HB10 strain. The survival rate of the cats challenged with the FCV-HB10 strain was higher than that of those given the FCV-HB7 strain. There were no obvious abnormalities in the DMEM control group (Figure 4B,C).

At the end of the observation period (14 dpi), all of the cats were euthanized and necropsied. Hematoxylin–eosin (H&E) staining showed (Figure 4A) that the cats in the FCV-HB7 and FCV-HB10 challenge groups had severe lesions in multiple tissues such as thickened alveolar walls, widened lung interstitium, increased red blood cell exudation, and most turbinate mucosa shedding and necrosis. No obvious tissue lesions were observed in the cats in the DMEM control group under the microscope. These results suggested that both the FCV-HB7 and FCV-HB10 strains were highly pathogenic to the tested cats, but the FCV-HB7 strain was more virulent than the FCV-HB10 strain (Figure 4A).

### 3.6. Cat Protective Efficacy of FCV Vaccine Strain and In Vitro Cross-Neutralization Test

To assess the immune efficacy of the FCV bivalent inactivated vaccine (FCV-HB7 and FCV-HB10), serum samples were collected on days 14 and 35 after the primary immunization (14 days after the booster immunization) to determine its antibody production period by measuring the neutralizing antibody titer. The results showed that the FCV bivalent inactivated vaccine induced a certain level of neutralizing antibody titer on day 14 after the initial immunization, which was ≥1:128, and it increased ≥1:256 on day 35 after the initial immunization (Figure 5B). On day 42 after the initial immunization (21 days after the booster immunization), the cats in the FCV bivalent inactivated vaccine immunization group and the DMEM control group were intranasally infected with the FCV-HB7 strain or FCV-HB10 strain at dose of 10^9.0^ TCID_50_/cat (Figure 5A). The results showed that the cats in the DMEM vaccination group developed severe typical clinical symptoms of FCV within 14 dpi, including weight loss, conjunctivitis, keratitis, mucopurulent eye and nose secreta, oral ulcers, and even death (Figure 5E,F), and substantial viral shedding occurred (Figure 5C). In contrast, the cats in the FCV bivalent inactivated vaccination group showed no obvious clinical symptoms, no death was observed during the 14-day observation period (Figure 5E,F), and the virus shedding was lower than that in the DMEM group (Figure 5C). Histopathological assay showed that the DMEM vaccination group had obvious severe microscopic lesions of multiple tissues such as the infiltration of eosinophilic protein-like substances, hemorrhage of alveolar cavities of the lungs, and most turbinate mucosa shedding and necrosis. However, the bivalent inactivated vaccine immunization group showed no obvious pathological changes (Figure 5D). In the viral load test, the viral load of each tissue in the DMEM group was also higher than that in the bivalent vaccine immunization group (Figure 5C).

In order to further evaluate the cross-protection of FCV vaccine strains, we determined the neutralizing antibody titers and in vitro cross-neutralizing antibody titers of the sera from cats immunized with each FCV vaccine strain (FCV-HB7, FCV-HB10, FCV-HB7 and FCV-HB10, FCV-255) against the FCV-representative strains. The results showed that the FCV-HB7 and FCV-HB10 vaccine strains produced extremely high neutralizing antibody titers after single-strain immunization, and both of them had certain cross-neutralizing titers against the other six representative FCV strains (Figure 5G, Appendix A). The neutralizing antibody titer of the FCV-HB7 vaccine against the FCV-HB7 strain after single-strain immunization was the highest, reaching 1:2^10.47^ (1:1414); its cross-neutralizing titer against the FCV-SH192 strain was the lowest at 1:2^5^ (1:32). After FCV-HB10 immunization, the neutralizing titer against FCV-HB10 was the highest, reaching 1:2^10.3^ (1:1260), and its cross-neutralizing titer against FCV-HB260 was the lowest at 1:2^4^ (1:16). After immunization with the bivalent inactivated vaccine FCV-HB7 and FCV-HB10, the neutralizing titers against the seven representative FCV strains were all at a high level, and none were lower than 1:2^7.2^ (1:148). However, after immunization with the commercial vaccine strain FCV-255, all of the neutralization titers against seven representative FCV strains were low, the maximum neutralization titer was only 1:2^5^ (1:32) against FCV-JL18, and the neutralization titers against the other six strains were below 1:2^2^ (1:4). These results showed that the FCV bivalent inactivated vaccine developed in this study had better in vitro cross-neutralizing antibody titers against current FCV epidemic representative strains than the current commercial vaccine.

## 4. Discussion

FCV is a highly mutagenic virus, exhibiting one of the highest evolution rates among known RNA viruses [15]. FCV strains exhibit high genetic plasticity, and no single prevalent strain dominates [31]. The evolution of viruses is not only based on competition among different isolates, but is also dependent on random mutations, reinfection, and recombination, resulting in long-term survival in susceptible populations or individuals. However, there still exist some genetic correlations among mutated strains, which enables cross-protection through vaccination [32].

Current viruses have evolved different strategies to evade the host’s immune response [32]. As a highly mutagenic RNA virus, FCV possesses an error-prone viral polymerase, and, thus, its genome continuously accumulates mutations, increasing the adaptability of the virus and eventually leading to immune evasion [32,33]. The continuous evolution of FCV viruses poses challenges to vaccine design [34]. Several studies have indicated that vaccine strains used over decades may have become less effective, and that there are differences in neutralizing titers between FCV epidemic isolates from different regions and vaccine strains, which has also been confirmed by our previous research [27,29]. The FCV-255 inactivated vaccine widely used in China has limited in vitro cross-neutralizing potency against the Chinese epidemic strains, and it may not provide effective protection [22,30,35]. Moreover, there are great differences in the structural proteins among the representative strains isolated from different regions in China in our previous study [29]. Based on the homology of the genes encoding capsid proteins and genetic evolutionary distance, we selected seven Chinese epidemic FCV strains as representative strains and conducted a series of biological characteristics studies to screen vaccine strains with broad cross-protection. This could potentially lead to the development of novel FCV vaccines especially suitable and effective for the Chinese market, thereby reducing the clinical incidence of the disease and allowing FCV to be better controlled.

In this study, the sequence lengths of seven representative FCV strains ranged from 7666 bp to 7706 bp. The whole-genome nucleotide homology between these seven strains and the reference strains was 75.00–90.30%, and that between these strains and the commercial vaccine strain FCV-255 was 77.37–80.60%, all of which were lower than 81.0%. The phylogenetic tree constructed based on the full-length sequences of FCV showed that the currently common vaccine strains belonged to groups A and B, while our seven isolates belonged to groups C and D with great genetic distance between them, which suggested that the currently used commercial FCV vaccine might have limited protective efficacy against the epidemic strains in China. Our results are similar to the findings of several previous studies in China, suggesting the diversity of epidemic FCV strains in the country [30,31,34,36].

We selected HB7 and HB10 from seven representative FCV strains as candidate vaccine strains by comparing the virus proliferation titers and the neutralizing antibody titers induced by inactivated antigen immunization. Further studies were conducted on the growth kinetics, pathogenicity in cats, and immune efficacy of these two strains. The results showed that the virus titers of the FCV-HB7 strain and FCV-HB10 strain reached the highest at 36 hpi at an inoculation dose of 0.01 MOI. This is consistent with the previous report that virus titers of FCV in vitro culture reached the peak at 24–36 hpi [30]. All of the cats intranasally infected with the FCV-HB7 strain or FCV-HB10 strain showed obvious characteristic clinical symptoms of FCV. The further histopathological assay showed severe lesions on the target tissues of the cats, which was in line with some previous reports [15,30]. Interestingly, the symptoms of the cats challenged with the FCV-HB7 strain were more severe than those of the cats challenged with the FCV-HB10 strain. The reason for the difference in virulence may be that three out of seven virulence-related characteristic amino acid sites of the FCV-HB7 strain were identical to those of the reported highly virulent FCV [29,37,38].

Passive immunity or natural infection with calicivirus can trigger an immune response, which has implications for the control of infection and viral transmission [21,39]. Some efforts have been made to develop an FCV vaccine producing broad humoral protection [15,37]. Homologous and heterologous FCV antibody responses are closely related to protection from FCV clinical disease [25]. Previous studies have demonstrated that neutralizing antibodies appeared in the cats from day 8 to day 14 after vaccination or infection, and homologous antibodies were detected earlier than heterologous antibodies [39,40]. Neutralizing antibody titers were significantly different among individuals [15]. In this study, we investigated the immune efficacy of the screened FCV-HB7 and FCV-HB10 strains through single-strain immunization and combined bivalent vaccine immunization. The results showed that both single-strain immunization and bivalent vaccine immunization could induce high neutralizing antibody titers with no significant difference between them. After combined bivalent inactivated vaccine immunization, the cats were protected from attacks by FCV-HB7 and FCV-HB10 strains and no obvious clinical symptoms or substantial viral shedding were demonstrated within 14 dpi, suggesting that the FCV bivalent inactivated vaccine we developed had good immune efficacy.

It has been reported that multiple factors should be taken into account in the development of broadly cross-protective vaccines against FCV, such as the high mutation rate of FCV different capsid protein antigen epitopes, host selection pressure, antibody cross-reactivity, long-lasting protection after infection, and the changing genome population every 5 years worldwide [15,32]. Cross-neutralization is an effective method for FCV vaccine strain selection and predicts significant protection against FCV, including reduced disease severity. This approach makes it possible to select FCV strains with broad cross-reactivity and develop a vaccine that provides protection against most circulating variants [28]. Considering this, we further investigated the broad cross-protection of the developed inactivated vaccine against epidemic strains in China after immunization. The results showed that single-strain immunization with FCV-HB7 or FCV-HB10 displayed certain cross-neutralizing titers against the other six FCV representative strains, but combined bivalent vaccine immunization exhibited higher cross-neutralizing titers than single-strain immunization. The cross-neutralizing titer of single-strain immunization and bivalent vaccine immunization was much higher than that of immunization with the commercial FCV-255 strain currently widely used in China. This is consistent with previous reports stating that current commercial vaccines may not provide effective protection against epidemic strains in China [22,35,41]. Previous reports have confirmed that the cross-reactivity of vaccines with the challenge strain is consistent with the heterologous protective effect of the vaccine [28]. The use of cross-neutralization to predict vaccine efficacy has been validated, and multivalent vaccines are better able to exert cross-protective effects [28]. Therefore, the development and cross-neutralization of the bivalent inactivated vaccine conducted in this study are necessary for the prevention and control of FCV in China. Previous reports investigating cases of upper respiratory tract disease indicate that cats fully immunized with the feline triple vaccine are still at a high risk of FCV with stomatitis at a 40.2% infection rate in China [39]. This further demonstrates that FCV is widely prevalent in China and is the main pathogen causing upper respiratory tract diseases. Developing vaccines with broader cross-protective effects against wild strains prevalent in China has great application prospects and economic value, which can reduce the prevalence of FCV to a certain extent. The results of this study indicate that the FCV-HB7 and FCV-HB10 strains screened have significant potential to become vaccine strains with broad-spectrum protective efficacy. If widely used, this vaccine is expected to effectively control the epidemic of FCV in China, providing a certain level of protection to China’s more than 60 million cats and reducing the clinical diseases caused by FCV infection. However, their immune protection efficacy needs to be further verified using multiple methods before clinical application as vaccine products.

## 5. Conclusions

In this study, FCV was divided into five groups, A, B, C, D, and E, based on the whole-genome sequence. The Chinese isolates belonged to groups C and D. The preliminarily screened FCV-HB7 and FCV-HB10 strains exhibited good in vitro culture properties, immunogenicity, and broad cross-protection. In vivo challenge experiments demonstrated that, as bivalent vaccine strains, the FCV-HB7 and FCV-HB10 strains had desirable protective efficacy against the group C and D strains epidemic in China after immunization.

## Figures and Tables

**Figure 1 vaccines-11-01858-f001:**
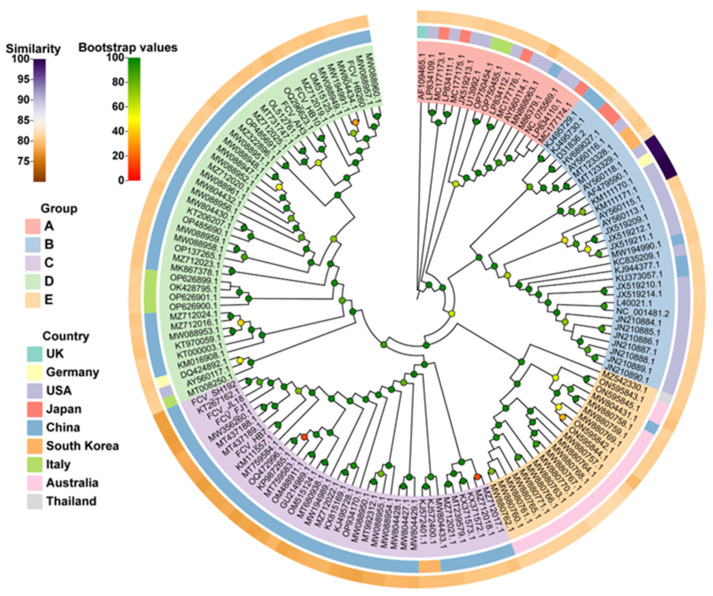
Whole-genome nucleotide sequence-based phylogenetic tree of seven FCV isolates and reference strains obtained from GenBank. A phylogenetic tree was constructed via the Maximum Likelihood (ML) method using the Kimura two-parameter model in the MEGA x software (11.0.13, Pennsylvania State University, State College, PA, USA) package with 1000 bootstrap replicates. The branches are marked with the GenBank accession number of each reference strain. Different colors of the outermost circle represent different levels of nucleotide homology (similarity) between other strains and FCV-255 vaccine strains. Different colors in the middle circle indicate different countries. Different colors in the innermost circle mark different groups. Bootstrap values are marked in different colors at each clade node.

**Figure 2 vaccines-11-01858-f002:**
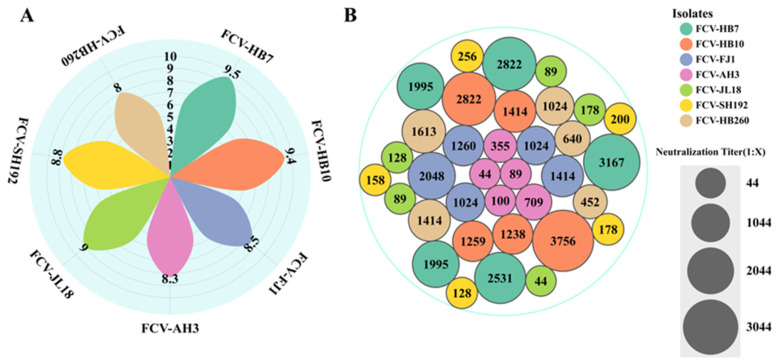
Virus titer and immunogenicity of 7 FCV isolates. (**A**) Virus titers when complete CPE was observed after inoculating 7 FCV strains into F81 cells. (**B**) Serum neutralizing antibody titers after immunization of SPF mice with 7 FCV isolates.

**Figure 3 vaccines-11-01858-f003:**
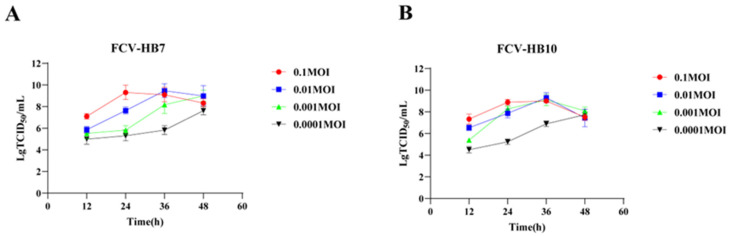
Growth kinetics of FCV vaccine strains. (**A**) Replication ability of FCV-HB7 strain at different inoculation doses. (**B**) Replication ability of FCV-HB10 strain at different inoculation doses.

**Figure 4 vaccines-11-01858-f004:**
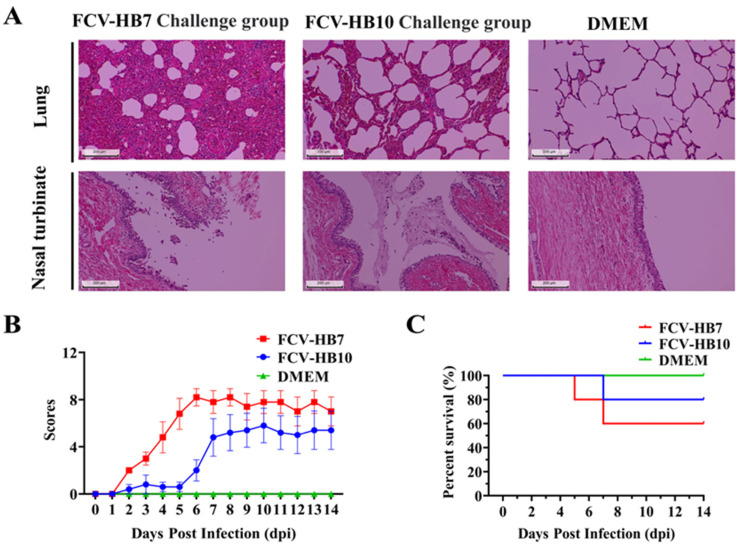
Health status observation of cats challenged with FCV-HB7 and FCV-HB10 strains. (**A**) HE staining of lung tissues and nasal turbinate of tested cats in each group. (**B**) Clinical symptom scores of tested cats within 14 dpi. (**C**) Survival curve of cats within 14 dpi.

**Figure 5 vaccines-11-01858-f005:**
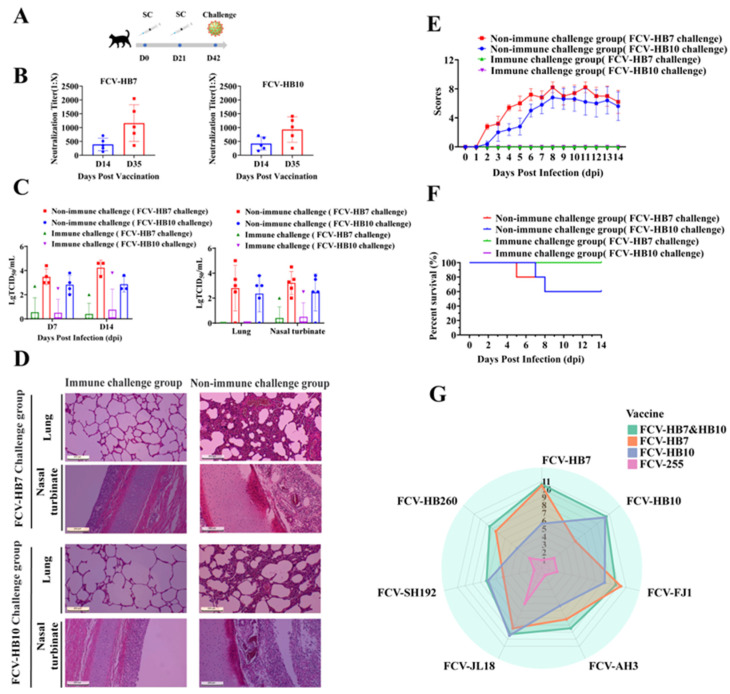
Cat protective efficacy experiment and in vitro cross-neutralization test. (**A**) Experimental design. (**B**) Neutralizing antibody titers on days 14 and 35 after initial immunization with FCV bivalent inactivated vaccine. (**C**) The TCID_50_ method was used to detect virus shedding in swabs on days 7 and 14 after infection, and viral load in lung tissue and nasal turbinate after death. (**D**) H&E staining of lungs and nasal turbinate of test cats in each group. (**E**) Cats’ clinical symptom scores within 14 days post infection. (**F**) Survival rates of cats in the immune-challenged and non-immune-challenged groups exposed to FCV-HB7 and FCV-HB10 strains. (**G**) Neutralizing antibody titers and in vitro cross-neutralizing antibody titers of cat sera after immunization with each FCV vaccine strain (FCV-HB7, FCV-HB10, FCV-HB7 and FCV-HB10, FCV-255). The radar radius represents the neutralizing antibody titer, expressed as a log_2_ value.

**Table 1 vaccines-11-01858-t001:** PCR primers for nucleotide amplification of FCV whole-genome sequence.

Primer	Sequences (5′−3′)	Tm (°C)	Product (bp)
FCV-Q1	F-GTAAAAGAAATTTGAGACAATGT	51	1–2468
	R-GAATTAACRGTTACCACATG		
FCV-Q2	F-GAACTACCCGCCAATCAACATGT	63	2431–5362
	R-AGCACGTTAGCGCAGGTTGAGC		
FCV-Q3	F-TTGAGCATGTGCTCAACCTG	55	5330–7684
	R-CCCTGGGGTTAGRCGC		

**Table 2 vaccines-11-01858-t002:** Animal clinical symptom scores.

Clinical Symptoms	Description of Clinical Symptoms	Score
Body temperature	37.5–39.5	0
	≥39.6	1
	≤37.4	2
Oral ulcers	None	0
	Small and few	1
	Big and many	3
Eye and nose secreta	None	0
	Few	1
	Many	2
Body weight	Increase or decrease <3%	0
	Reduce ≥3%	2
State of existence	Survival	0
	Death	10

**Table 3 vaccines-11-01858-t003:** Seven FCV isolates identified in this study.

Isolated	Time	Area	Group	Length (bp)	GenBank No.
FCV-HB7	2020	Hubei, China	C	7683	OR645484
FCV-HB10	2020	Hubei, China	D	7706	OR645485
FCV-FJ1	2021	Fujian, China	C	7683	OR645479
FCV-AH3	2021	Anhui, China	D	7706	OR645480
FCV-JL18	2021	Jilin, China	C	7683	OR645481
FCV-SH192	2021	Shanghai, China	C	7666	OR645482
FCV-HB260	2021	Hubei, China	D	7704	OR645483

**Table 4 vaccines-11-01858-t004:** Sequence homology of three open reading frames (ORFs) between feline calicivirus (FCV) isolates and FCV 255 strain.

Strains	*ORF1*		*ORF2*		*ORF3*	
	*nt*	*aa*	*nt*	*aa*	*nt*	*aa*
FCV-HB7	76.02	87.65	74.54	83.53	78.19	85.85
FCV-HB10	78.61	91.16	77.11	88.04	82.87	94.34
FCV-FJ1	76.26	87.14	74.54	84.58	77.26	83.96
FCV-AH3	78.80	91.10	77.21	87.00	85.36	94.34
FCV-JL18	75.54	86.97	74.99	85.03	78.19	84.91
FCV-SH192	76.69	87.25	75.04	86.92	78.82	83.18
FCV-HB260	78.99	91.10	78.34	87.28	83.54	95.28

## Data Availability

The data used to support the findings of this study are included in this article.

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
