# Peer review of "Biological Characteristics of Feline Calicivirus Epidemic Strains in China and Screening of Broad-Spectrum Protective Vaccine Strains"

_vaccines, 2023, doi:10.3390/vaccines11121858_

Round 1
Reviewer 1 Report
Comments and Suggestions for Authors
The FCV bivalent inactivated vaccine prepared by the institute is expected to effectively control the epidemic of FCV in China. Taking into account the multiple studies ongoing in this field this type of study is needed. I have few minor remarks that authors should address properly.
The following minor revisions are recommended:
1. Line 99: Add strain GenBank No.
2. Line 106: Seven strains of FCV are used as representative strains from different regions in China. Is there any background information?
3. FCV is often divided into genotype I and genotype II. However, in Fig.1 and 3.2 of this study, FCV is divided into groups A, B, C, D, and E. What is the basis for the grouping? Are there any literature references?
4. The strains isolated in this study should be labeled in Fig.1; the "Group" in the legend should be labeled in the order of ABCDE; the phylogenetic tree constructed appears to be a rooted tree? Neighbor-joining cannot construct such a rooted tree. Please determine if the method is wrong and correct it.
5. Line 287, Line 332: The format of all groups of figures in Fig.4 and Fig.5 should be consistent, for example, the y-axis font should be in the same direction.
6. Some language tenses are not used rigorously in the article. It is recommended to check.
For example:
Line 369, "all of which was lower than 81.0%" should be as "all of which were lower than 81.0%";
Line 370, "full-length sequence of FCV" change into "full-length sequences of FCV";
Line 389, "was" Should be written as "were";
Line 390, "can trigger a humoral response" revised to "can trigger a humoral immune response".
Comments on the Quality of English LanguageMinor editing of English language required
Author Response
Dear Reviewer,
On behalf of my co-authors, we thank you very much for giving us an opportunity to revise our manuscript, we appreciate editor and reviewers very much for their positive and constructive comments and suggestions on our manuscript entitled “Biological characteristics of feline calicivirus epidemic strains in China and screening of broad-spectrum protective vaccine strains” (MS ID: vaccines-2701332).
We have carefully read reviewers’ comments and have made revision which marked in blue in the paper. We have tried our best to revise our manuscript according to the comments. Below please find our responses to the reviewers’ comments point-by-point.
We would like to express our great appreciation to you and reviewers for comments on our paper.
Looking forward to hearing from you.
Thank you and best regards.
Yours sincerely,
Shengbo Cao, PhD.
Corresponding author.
E-mail: [email protected]
Responses to the comments:
1) Line 99: Add strain GenBank No.
Response: Thanks for the comment. According to the Reviewer’s suggestion, GenBank No. of the strain has been added in 2.1Cell and virus. Please check lines 93-98.
2) Line 106: Seven strains of FCV are used as representative strains from different regions in China. Is there any background information?
Response: Thank you for your suggestions. We have provided background information in 2.1Cell and virus. Please check lines 93-98.
3) FCV is often divided into genotype I and genotype II. However, in Fig.1 and 3.2 of this study, FCV is divided into groups A, B, C, D, and E. What is the basis for the grouping? Are there any literature references?
Response: Thank you for your comment. At present, there is no clear standard for the basis of FCV whole genome sequence grouping. According to the study of Mao J et al (https://doi.org/10.3390/v14112421), FCV can be divided into four genogroups (ABCD). Based on the whole genome sequences, and in this study, we added the whole genome sequences of atypical FCV strains, and these strains independently formed a topological group in the phylogenetic tree, so we tried to divide FCV into five genogroups, ABCDE, based on the whole genome sequences, and we hope that it can bring a clear message for the subsequent research on FCV. We hope that this will be helpful for subsequent FCV research.
4) The strains isolated in this study should be labeled in Fig.1; the "Group" in the legend should be labeled in the order of ABCDE; the phylogenetic tree constructed appears to be a rooted tree? Neighbor-joining cannot construct such a rooted tree. Please determine if the method is wrong and correct it.
Response: Thank you for the comment. The legend has been changed and the method modified. Please check Figure 1 and lines 114-115.
5) Line 287, Line 332: The format of all groups of figures in Fig.4 and Fig.5 should be consistent, for example, the y-axis font should be in the same direction.
Response: Thank you for the suggestions. Modifications have been made, please check Figure 4 and Figure 5.
6) Some language tenses are not used rigorously in the article. It is recommended to check.
For example:
Line 369, "all of which was lower than 81.0%" should be as "all of which were lower than 81.0%";
Line 370, "full-length sequence of FCV" change into "full-length sequences of FCV";
Line 389, "was" Should be written as "were";
Line 390, "can trigger a humoral response" revised to "can trigger a humoral immune response".
Response: Thank you for the suggestions. These errors have been corrected, and the full text has been checked and corrected. Please check lines 397, 398, 416, and 418.
Once again, we are grateful to your suggestions.
Reviewer 2 Report
Comments and Suggestions for Authors
After reading only the abstract and introduction my recommendation is not to publish this version of the paper. It is a problem with English language, but also with the merits of the matter.
Below some remarks concerning only the first page (and please find more in the PDF attached).
The basic question is: what does it mean “epidemic” strains? The term “epidemic strains” is commonly used in the text, but there is no explanation what does it mean.
line 26 “All cats intranasally infected with FCV-HB7 or FCV-HB10 strain showed obvious clinical symptoms of FCV. The symptoms of cats challenged with FCV-HB7 strain were more severe than those challenged with FCV-HB10 strain”. Please, specify how many cats?
line 24 “The screened two vaccine strains displayed desirable in vitro culture characteristics, with the...” is about vaccine strains, next 2 sentences (obviously?) about native strains, and then again about vaccine strains. Please put this mixture of strains in proper order .
line 27 „The symptoms of cats challenged with...” please replace “challenged” by “infected”
line 28 “Both single strain immunization and combined bivalent vaccine...” live attenuated or inactivated vaccine?
line 30 “bivalent inactivated vaccine immunization protected cats from attacks by FCV-HB7 and....” please replace “from attacks” by “challenged”
line 31 “Cross-neutralizing antibody titer achieved by combined bivalent vaccine...” cross-neutralization between which strains?
In general:
In the Introduction there are overinterpretations of the papers cited (see PDF of the paper).
The goal of the study is not clear (why 2 stains were compared on cats? what for? what was the hypothesis?).
In the abstract the numbers of cats are lacking, also information about the strains tested (origin? how many?) and type of vaccine (inactivated? adjuvanted? etc.).
In the Material and Methods section there is nothing about the origin of the strains studied (isolated from the respiratory tract? oral cavity? feces? virulent-systemic?) .

Comments on the Quality of English LanguageEnglish needs extensive improvement
Author Response
Dear Reviewer,
On behalf of my co-authors, we thank you very much for giving us an opportunity to revise our manuscript, we appreciate editor and reviewers very much for their positive and constructive comments and suggestions on our manuscript entitled “Biological characteristics of feline calicivirus epidemic strains in China and screening of broad-spectrum protective vaccine strains” (MS ID: vaccines-2701332).
We carefully considered the reviewers’ comments and revised them, with extensive editing for the English language. Modifications are marked in blue in the paper. We have tried our best to revise our manuscript according to the comments. Below please find our responses to the reviewers’ comments point-by-point.
We would like to express our great appreciation to you and reviewers for comments on our paper.
Looking forward to hearing from you.
Thank you and best regards.
Yours sincerely,
Shengbo Cao, PhD.
Corresponding author.
E-mail: [email protected]
Responses to the comments:
1) Below some remarks concerning only the first page (and please find more in the PDF attached).
Response: We are grateful to your good comments and suggestions. We have revised them item by item based on the issues raised in the PDF attachment and marked them in blue in the new manuscript. Please check lines 17-21, 25-27, 44-47, 56-60, 63-68, and 72-85.
2) The basic question is: what does it mean “epidemic” strains? The term “epidemic strains” is commonly used in the text, but there is no explanation what does it mean.
Response: Thanks for your comment. We provide an appropriate explanation of “epidemic strains” in the Abstract and Preface. Please check lines 17-21 and 81-83.
3) line 26 “All cats intranasally infected with FCV-HB7 or FCV-HB10 strain showed obvious clinical symptoms of FCV. The symptoms of cats challenged with FCV-HB7 strain were more severe than those challenged with FCV-HB10 strain”. Please, specify how many cats?
Response: Thank reviewer’s comments and good suggestion. “All cats intranasally infected with FCV-HB7 or FCV-HB10 strain showed obvious clinical symptoms of FCV…” has been modified to “All five cats infected intranasally with FCV-HB7 or FCV-HB10 strains showed obvious clinical symptoms of FCV…” Please check lines 27-28.
4) line 24 “The screened two vaccine strains displayed desirable in vitro culture characteristics, with the...” is about vaccine strains, next 2 sentences (obviously?) about native strains, and then again about vaccine strains. Please put this mixture of strains in proper order .
Response: Thank reviewer’s comments and good suggestion. “The screened two vaccine strains displayed desirable in vitro culture characteristics, with the...” has been modified to “The screened FCV-HB7 and FCV-HB10 strains displayed desirable in vitro culture characteristics, with the…” Please check lines 25-27.
5) line 27 “The symptoms of cats challenged with...” please replace “challenged” by “infected”
Response: Thanks for your suggestion, “challenged” has been changed to “infected”. Please check line 29.
6) line 28 “Both single strain immunization and combined bivalent vaccine...” live attenuated or inactivated vaccine?
Response: Thank you for your comment. It is an inactivated vaccine. “Both single strain immunization and combined bivalent vaccine...” has been modified to “Both single strain inactivated immunization and combined bivalent inactivated vaccine…”. Please check lines 29-30.
7) line 30 “bivalent inactivated vaccine immunization protected cats from attacks by FCV-HB7 and....” please replace “from attacks” by “challenged”
Response: We are grateful to your good comments and suggestions. “from attacks” has been replaced by “challenged”. Please check line 32.
8) line 31 “Cross-neutralizing antibody titer achieved by combined bivalent vaccine...” cross-neutralization between which strains?
Response: Thank you for your comment. I have added strain information. Please check line 33.
9) In general: In the Introduction there are overinterpretations of the papers cited (see PDF of the paper).
Response: Thank you for your comment. We have revised the over-interpretation of the cited papers in the introduction. Please check lines 75-80.
10) The goal of the study is not clear (why 2 stains were compared on cats? what for? what was the hypothesis?).
Response: Thank you for your comment. In this study, based on the differences in capsid proteins of isolates from different regions in China, as reported in our previous studies, 7 representative FCV epidemic strains were selected and tested for their viral titers, virulence, immunogenicity, and extensive cross-protection. Subsequently, vaccine strains were selected to prepare inactivated vaccines. In order to provide better cross-protection against FCV strains prevalent in different regions of China. The two strains were compared in cats because the two FCV strains showed the best immunogenicity after immunizing mice, but their use as vaccine strains should be verified on target animals. The prerequisite for vaccine efficacy verification is to establish an evaluation system based on animal models. Therefore, its pathogenicity and efficacy in cats were verified. To further verify the hypothesis that it can become a broad-spectrum cross-protective vaccine strain.
11) In the abstract the numbers of cats are lacking, also information about the strains tested (origin? how many?) and type of vaccine (inactivated? adjuvanted? etc.).
Response: Thank you for your comment. I have added to the abstract information about the number of cats in each trial, the strains tested, and clarified that the type of vaccine was an adjuvanted, inactivated vaccine. Please check lines 27-31.
12) In the Material and Methods section there is nothing about the origin of the strains studied (isolated from the respiratory tract? oral cavity? feces? virulent-systemic?).
Response: Thank you for your suggestions. We have provided background information in 2.1 Cell and virus. Please check lines 93-98.
Once again, we are grateful to your suggestions.
Reviewer 3 Report
Comments and Suggestions for Authors
The authors described the process of selecting FCV vaccine strains to develop a bivalent FCV inactivated vaccine. Overall, the article is well written and organized. The design of some figures is original but always the most appropriate to convey a simple information. The main limitation of this study is the FCV titer of the inoculum in the cat studies. It is not at all representative of the natural conditions of infection (cat never excrete 10^9 TCID50) and it may impact some of the conclusions of the article. Another limitation is the absence of shedding data, which is a critical parameter in the ability of vaccination to control the circulation of the virus in the cat population. Importantly, the authors must describe how they made sure that the virus was fully inactivated. Reviewer’s comments are below:
Line 43: Pneumonia is not a usual clinical sign of classical FCV strains. It may be a clinical sign in cats infected with hypervirulent strains.
Lines 61-63: The authors should document this sentence and provide references.
Lines 114-120: how did the authors check that the virus was fully inactivated? With beta-propiolactone, you may have residual live virus and a sensitive virus detection test is needed to check full inactivation.
Lines 156-157: the titer of the inoculum is very high. Usually, challenge experiments with FCV are done with 10^6 TCID50. The authors should comment on this.
Figure 1: it is difficult to see the colors of the bootstrap values in the small circles of the figure.
Lines 250-254: it looks like the vaccine strains were selected on the basis of homologous neutralizing titers in mice. Could the authors elaborate on the rationale behind this selection process?
Figure 3: classical titer curves would have made it easier to read. This type of figure is not the most appropriate for representing viral growth.
Lines 271-277: it is surprising to see death after FCV infection, unless a hypervirulent FCV strain is used. Clinical signs other than death are those of classical FCV strains. One wonder whether the deaths were not caused by the very high viral inoculum titer. This raises some questions on the representativity of the field conditions of infection. The authors should comment on this.
Line 290 (section 3.6): Is the vaccine adjuvanted? If it is, what is the adjuvant?
Lines 311-330: the authors should explain how they performed the cross-neutralization study. Did they use the serum of a single cat? In that case, how was the cat selected? Did they pool sera from different cats? How many? In addition, a table with all titers (homologous and heterologous) would be a valuable addition.
Lines 387-389: as previously written, assumptions around the pathogenicity of FCV strains HB7 and HB10 are very speculative because the titer of the inoculum administered to cats is extremely high and not representative of the infectious dose cats may receive in natural infection. The authors should consider this as a limitation of their study.
Lines 390-405: in relation with the question on the vaccine formulation, the authors should comment on the antibody titers after vaccination. If the vaccine was adjuvanted, these antibody titers are consistent. If the vaccine is not adjuvanted, the authors should explain how a non-adjuvanted inactivated vaccine can generate such high antibody titers. The description of the inactivation control test would be useful. With beta-propiolactone, one may have residual live virus. An incomplete inactivation could contribute to higher immune response.
Lines 423-425: the authors did not study the impact of vaccination against FCV shedding after infectious challenge. It would have been a valuable piece of information especially if we expect the vaccine to contribute to a reduction of the circulation of FCV strains in the cat population. The authors should comment on why this parameter was not studied. Same remark applies to virus isolation from damaged lungs. It would have been a useful information to measure the viral load in the lungs.
Lines 429-430: it seems a little odd to speak of economic loss for a disease affecting companion animals. Could the authors elaborate on what they call economical loss?
Discussion: the author’s approach is the same as that described in Poulet et al., Vaccine 2008 (doi: 10.1016/j.vaccine.2008.04.082). It might be interesting to compare the outcome of the studies. It might reinforce the concept of bivalent vaccine.
References 13 and 35 are the same article.
Comments on the Quality of English LanguageQuality of English language is good overall.
Author Response
Dear Reviewer,
On behalf of my co-authors, we thank you very much for giving us an opportunity to revise our manuscript, we appreciate editor and reviewers very much for their positive and constructive comments and suggestions on our manuscript entitled “Biological characteristics of feline calicivirus epidemic strains in China and screening of broad-spectrum protective vaccine strains” (MS ID: vaccines-2701332).
We have carefully read reviewers’ comments and have made revision which marked in blue in the paper. We have tried our best to revise our manuscript according to the comments. Below please find our responses to the reviewers’ comments point-by-point.
We would like to express our great appreciation to you and reviewers for comments on our paper.
Looking forward to hearing from you.
Thank you and best regards.
Yours sincerely,
Shengbo Cao, PhD.
Corresponding author.
E-mail: [email protected]
Responses to the comments:
1) Line 43: Pneumonia is not a usual clinical sign of classical FCV strains. It may be a clinical sign in cats infected with hypervirulent strains.
Response: Thanks for the comment. According to the Reviewer’s suggestion, revised to “…and it mainly causes symptoms such as oral ulcers and respiratory, tracheitis in felines…”. Please check lines 44-45.
2) Lines 61-63: The authors should document this sentence and provide references.
Response: Thanks for your comment. Reference [24] has been added. Please check lines 62-65.
3) Lines 114-120: how did the authors check that the virus was fully inactivated? With beta-propiolactone, you may have residual live virus and a sensitive virus detection test is needed to check full inactivation.
Response: Thank reviewer’s comments. In the inactivation test, the specific method involves adding β-propiolactone and shaking at 4°C for 24 hours. After that, the β-propiolactone is hydrolyzed at 37°C for 2 h. Once inactivation is complete, the antigen is inoculated into the F81 cell culture medium covering the T25 monolayer at a volume ratio of 1:100. Daily observation of lesions is conducted. If no lesions are observed, harvesting is performed by freezing and thawing on the 6th day. Blind passage is then performed for 3 generations using the same inoculation method. If no lesions are observed in the 3 generations, the inactivation process is considered successful. Please check lines 131-138.
4) Lines 156-157: the titer of the inoculum is very high. Usually, challenge experiments with FCV are done with 10^6 TCID50. The authors should comment on this.
Response: Thank reviewer’s comments and good suggestion. Although the 109.0 TCID50 virus titer we used cannot represent the natural conditions of infection, the animal experimental model established at this dose can better evaluate the protective effect of subsequent vaccine immunization.
5) Figure 1: it is difficult to see the colors of the bootstrap values in the small circles of the figure.
Response: Thanks for your suggestion. Figure 1 has been modified, changed into intuitive colors, and the circles have been enlarged. Please check Figure 1.
6) Lines 250-254: it looks like the vaccine strains were selected on the basis of homologous neutralizing titers in mice. Could the authors elaborate on the rationale behind this selection process?
Response: Thank you for your comment. Mouse models are widely used for immunogenicity assessment, primarily because the mouse immune system is structurally and functionally similar to the cat immune system, providing several advantages. Firstly, the immune system of mouse models is highly conserved. Many basic immune response mechanisms are shared between mice and cats, including antigen presentation, antibody production, cellular immunity, etc. Assessing the immune response in mice provides valuable information about the immune response in cats. Secondly, mouse models offer highly controlled experimental conditions. Moreover, mouse models are relatively cost-effective and easy to operate. Lastly, ethical and legal considerations are also factors in choosing mouse models. While mouse models offer several advantages in immunogenicity assessment, results still need to be interpreted and generalized with caution. Since mouse models only represent specific aspects of the cat immune system. Hence, when screening vaccine strains, we conducted additional clinical trials on cats to verify the immunogenicity and effectiveness of the vaccine.
7) Figure 3: classical titer curves would have made it easier to read. This type of figure is not the most appropriate for representing viral growth.
Response: We are grateful to your good comments and suggestions. It has been modified to classic titer curves. Please check Figure 3.
8) Lines 271-277: it is surprising to see death after FCV infection, unless a hypervirulent FCV strain is used. Clinical signs other than death are those of classical FCV strains. One wonder whether the deaths were not caused by the very high viral inoculum titer. This raises some questions on the representativity of the field conditions of infection. The authors should comment on this.
Response: Thank you for your comment. After being infected with FCV, in addition to severe stomatitis, the test cats also had a large amount of eye and nose secretions and respiratory symptoms, which greatly affected the test cats’ diet, which may be the main cause of death. Secondly, as the clinical symptoms after FCV infection worsen, it may also cause secondary self-trapping disease, which may also be one of the causes of death.
9) Line 290 (section 3.6): Is the vaccine adjuvanted? If it is, what is the adjuvant?
Response: Thank you for your comment. Vaccines are adjuvanted. Mix the inactivated antigen and MONTANIDE™ GEL 02 PR (Seppic, France) adjuvant at a volume ratio of 9:1 to prepare the vaccine. Please check lines 138-140.
10) Lines 311-330: the authors should explain how they performed the cross-neutralization study. Did they use the serum of a single cat? In that case, how was the cat selected? Did they pool sera from different cats? How many? In addition, a table with all titers (homologous and heterologous) would be a valuable addition.
Response: Thank you for your comment. When performing a cross-neutralization study, serum from a randomly selected cat in each group is used. One completely immunized serum was selected from each group for cross-neutralization studies against other strains. Please check lines 219-222.
A table of all titers (homologous and heterologous) has been supplemented in Table S1.
Table S1. Neutralizing antibody titers and in vitro cross-neutralizating antibody titers of cat sera after immunization with each FCV vaccine strain ((FCV-HB7, FCV-HB10, FCV-HB7&FCV-HB10, FCV-255).
Group |
FCV-HB7 |
FCV-HB10 |
FCV-FJ1 |
FCV-AH3 |
FCV-JL18 |
FCV-SH192 |
FCV-HB7&HB10 |
10.63 |
10.47 |
9.64 |
8.47 |
9.30 |
7.20 |
FCV-HB7 |
10.47 |
5.46 |
10.30 |
7.20 |
8.47 |
5.00 |
FCV-HB10 |
5.64 |
10.30 |
8.20 |
5.29 |
9.47 |
7.00 |
FCV-255 |
1.00 |
2.00 |
2.00 |
1.00 |
5.00 |
1.00 |
Note: neutralizing antibody titers are expressed as 1:log2 value.
11) Lines 387-389: as previously written, assumptions around the pathogenicity of FCV strains HB7 and HB10 are very speculative because the titer of the inoculum administered to cats is extremely high and not representative of the infectious dose cats may receive in natural infection. The authors should consider this as a limitation of their study.
Response: Thank you for your comment. The hypothesis that FCV-HB7 is more pathogenic than FCV-HB10, mentioned in the article, is based on the fact that in our previous study (doi:10.1186/s44149-022-00047-7), the amino acid residues of HB7 at 7 reported virulence-related sites were closely related to those of highly virulent viruses. Three strains were identical, which differed from HB10.The difference in clinical symptoms after HB7 and HB10 infected test cats with the same virus titer led us to draw this hypothesis. However, this is only our hypothesis, and further research is needed to verify it.
12) Lines 390-405: in relation with the question on the vaccine formulation, the authors should comment on the antibody titers after vaccination. If the vaccine was adjuvanted, these antibody titers are consistent. If the vaccine is not adjuvanted, the authors should explain how a non-adjuvanted inactivated vaccine can generate such high antibody titers. The description of the inactivation control test would be useful. With beta-propiolactone, one may have residual live virus. An incomplete inactivation could contribute to higher immune response.
Response: We are grateful to your good comments and suggestions. The vaccine contains an adjuvant, and the formula is 9:1 between antigen and adjuvant. In the inactivation test, the specific method involves adding β-propiolactone and shaking at 4°C for 24 hours. After that, the β-propiolactone is hydrolyzed at 37°C for 2 h. Once inactivation is complete, the antigen is inoculated into the F81 cell culture medium covering the T25 monolayer at a volume ratio of 1:100. Daily observation of lesions is conducted. If no lesions are observed, harvesting is performed by freezing and thawing on the 6th day. Blind passage is then performed for 3 generations using the same inoculation method. If no lesions are observed in the 3 generations, the inactivation process is considered successful. Please check lines 131-138.
13) Lines 423-425: the authors did not study the impact of vaccination against FCV shedding after infectious challenge. It would have been a valuable piece of information especially if we expect the vaccine to contribute to a reduction of the circulation of FCV strains in the cat population. The authors should comment on why this parameter was not studied. Same remark applies to virus isolation from damaged lungs. It would have been a useful information to measure the viral load in the lungs.
Response: Thank you for your suggestion. This set of parameters is crucial; please forgive our oversight. We have added the virus shedding data to Figure 5C. We used the TCID50 method to detect virus shedding in eye and nasopharyngeal swabs on days 7 and 14 after infection. Among them, the viral load shed by the non-immune challenge group was higher than that of the immune challenge group, and the same trend was also observed in the tissues.
14) Lines 429-430: it seems a little odd to speak of economic loss for a disease affecting companion animals. Could the authors elaborate on what they call economical loss?
Response: Thank you for your valuable comments. We consider the economic losses associated with treating and alleviating clinical symptoms resulting from FCV infection. However, our consideration may be inadequate. We have removed the discussion on economic losses here and focused solely on the reduction of clinical diseases caused by FCV infection. Please check lines 465-467.
15) Discussion: the author’s approach is the same as that described in Poulet et al., Vaccine 2008 (doi: 10.1016/j.vaccine.2008.04.082). It might be interesting to compare the outcome of the studies. It might reinforce the concept of bivalent vaccine.
Response: We are grateful to your good comments and suggestions. This is a very valuable article and we have read it carefully and have quoted it in the text. The relevant results were compared and discussed. Please check lines 80-81, 438-442, and 451-454.
16) References 13 and 35 are the same article.
Response: Thank you for your comment. Please forgive our mistake, I have replaced reference 13, Please check lines 528-530.
Once again, we are grateful to your suggestions.